# An Orthogonal Type Two-Axis Lloyd's Mirror for Holographic Fabrication of Two-Dimensional Planar Scale Gratings with Large Area

**Xinghui Li [1,*,†], Haiou Lu [1,†], Qian Zhou [1,*], Guanhao Wu [1,2], Kai Ni [1] and Xiaohao Wang [1]**

1  Division of Advanced Manufacturing, Graduate School at Shenzhen, Tsinghua University, Tsinghua Campus, Xili University Town, Shenzhen 518055, China; lho16@mails.tsinghua.edu.cn (H.L.); guanhaowu@tsinghua.edu.cn (G.W.); ni.kai@sz.tsinghua.edu.cn (K.N.); wang.xiaohao@sz.tsinghua.edu.cn (X.W.)
2  Department of Precision Instrument, Tsinghua University, Haidian District, Beijing 100084, China
*  Correspondence: li.xinghui@sz.tsinghua.edu.cn (X.L.); zhou.qian@sz.tsinghua.edu.cn (Q.Z.)
†  These authors contributed equally to this work.

**Abstract:** In this paper, an orthogonal type two-axis Lloyd's mirror interference lithography technique was employed to fabricate two-dimensional planar scale gratings for surface encoder application. The two-axis Lloyd's mirror interferometer is composed of a substrate and two reflective mirrors (X- and Y-mirrors), which are placed edge by edge perpendicularly. An expanded and collimated beam was divided into three beams by this interferometer, a direct beam and two reflected beams, projected onto the substrate, X- and Y-mirrors, respectively. The unexpected beam sections having twice reflected off the mirrors were blocked by a filter. The remaining two reflected beams interfered with the direct beam on the substrate, generating perpendicularly cross patterns thus forming two-dimensional scale gratings. However, the two reflected beams undesirably interfere with each other and generate a grating pattern along 45-degree direction against the two orthogonal direction, which influence the pattern uniformity. Though an undesired grating pattern can be eliminated by polarization modulation with introduction of waveplates, spatial configuration of waveplates inevitably downsized the eventual grating, which is a key parameter for grating interferometry application. For solving this problem, theoretical and experimental study was carefully carried out to evaluate the fabrication quality with and without polarization modulation. Two-dimensional scale gratings with a 1 µm period in X- and Y-directions were achieved by using the constructed experiment system with a 442 nm He-Cd laser source. Atomic force microscopy (AFM) images and the result of diffraction performances demonstrated that the orthogonal type two-axis Lloyd's mirror interferometer can stand a small order undesired interference, that is, a degree of orthogonality between two reflected beams, denoted by $\gamma$, no larger than a nominal value of 0.1.

**Keywords:** interference lithography; two-axis planar scale grating; Lloyd's mirror; surface encoder

---

## 1. Introduction

Planar encoders play a key role in precision positioning of linear stages due to their high resolution, high robustness, and relatively low cost [1–9]. The key component in the planar encoder is a two-dimensional (2D) planar scale grating, on which typically surface relief square holes or sinusoidal hills with a period spacing from submicron to several microns are arrayed along two orthogonal directions.

Laser interference lithography (LIL), engine machining, and imprinting are three typical methods for fabrication of the scale grating. Compared with engine machining or imprinting, LIL is more

convenient and cost-effective because only a coherent laser source is required and neither sophisticated precision positioning system nor expensive photolithography equipment and mask preparation are required. For fabrication of a 2D planar grating, according to the exposure times, the LIL process can be separated into two categories, double exposure and single exposure. For the two-beam two-exposure, it requires twice exposure procedures and the second time exposure requires rotation of the substrate by 90 degrees [5,10]. The exposure can be based on an amplitude-division type two-beam LIL, called two beam two exposures (TBTE) or a wavefront-division type Lloyd's mirror type LIL, called one beam two exposures (OBTE) [11]. TBTE is good at fabrication of gratings with a relatively large area up to 100 mm × 100 mm, while OBTE can be easily handled with a relatively simple and stable optical system. However, a big challenge existing in the method of the two-exposure process is that the grating structures generated in the first exposure will be influenced by the background light in the second exposure. Thus the depths of the grating structures will be different in two orthogonal directions. As a result, diffraction efficiencies of both the negative and positive diffraction beams will not be consistent. This is a critical problem for the encoder system since it directly affects the accuracy and the detecting limit of the interference signals. Therefore, efforts for solving this problem using a single exposure for 2D scale grating have been done these years. There are two trends for this motivation, multibeam LIL and one beam LIL. Multibeam LIL includes, but is not limited to, triple-beam and quadruple-beam interference technique. This technology can achieve the goal of 2D grating patterns with a single exposure. Some researchers invented a novel nanofabrication technique, called four-beam interference lithography, by using a two-dimensional grating [12]. Nevertheless, it cannot be neglected that the multibeam interference lithography has complicated optical structure easily to be disturbed by external environment. Furthermore, the optical path difference (OPD) is also difficult to be manipulated.

In contrast, one beam LIL, represented by two-axis Lloyd's mirror interferometer and a corner-like reflector interferometer, allows fabricating a crossed 2D grating with a single exposure [13–18]. For one beam LIL, one more proposal called the two-axis Lloyd's mirror interferometer with polarization modulation technique, proposed in 2014, has the advantages of large grating area, easy handling, and stable grating patterns [13]. However, optimization of the theoretical inclination in grating shape requires sophisticated in situ detection system for exposure and development. The inclination will influence the diffraction efficiency in a certain degree when it is directly used as a photoresist grating, though this influence might be reduced when a further etched process. On the other hand, the corner-like reflector interferometer, also called orthogonal type two-axis Lloyd's mirror interferometer, also proposed in 2014, has the advantages on grating shape in depth direction, although the complicated interference patterns on the grating substrate require deep modification for application as a 2D scale grating [14]. For modification of the grating patterns, Yuki and Chen took polarization modulation technique and results verified this technology has a great potential for 2D scale grating fabrication. But it should be noted that, introduction of the polarization modulation both reduce the grating area and bring a new challenge in the system stability [16–18].

Thus, investigation of the orthogonal type two-axis Lloyd's mirror interferometer based LIL without polarization modulation for a large grating area is carried out in this research. We constructed an orthogonal type two-axis Lloyd's mirror interferometer and fabricated 2D scale grating under different polarization state. Design details, construction, experimental setup, and evaluation are presented in this paper.

## 2. Methods

### 2.1. Optical Setup of the Exposure System

An orthogonal two-axis Lloyd's mirror interferometer was designed and assembled, as shown in Figure 1, which mainly includes a light source, beam shaping unit, polarization adjustment unit, and two-axis Lloyd's mirror unit. For the light source, with consideration of laser source quality (stability of power, polarization, and wavelength), photoresist sensitivity against laser wavelength, and optical

alignment low-complexity related to laser visibility, we employed a commercial, well-used He-Cd laser of the KIMMON KOHA company (Tokyo, Japan) with the wavelength of 442 nm and power output of 180 mW. Two optical elements are contained in the beam-shaping unit: the spatial filter element and collimating lens element. The two-axis Lloyd's mirror unit, as shown in Figure 2a, consists of three elements: X-mirror, Y-mirror, and the jig of substrate. Besides, the three elements are placed perpendicularly to each other. In addition, the two-axis Lloyd's mirror unit possess the function of rotation so that it's easy to manipulate the grating period. We designed the azimuth angle $\varphi$ equals to 45°, and the angle $\theta$ (0–90°) called the incident angle in this paper.

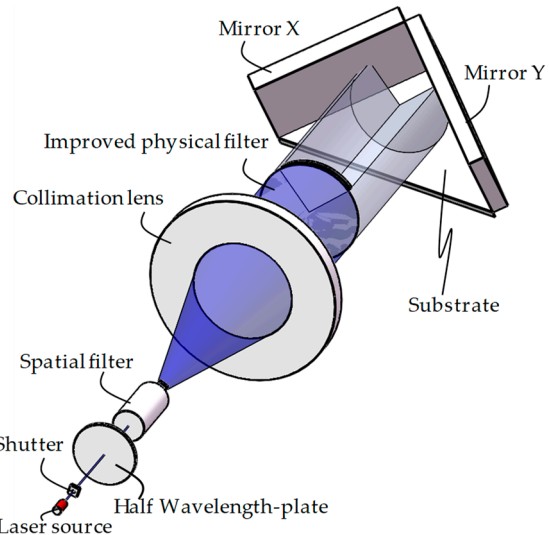

**Figure 1.** The optical configuration of the orthogonal type two-axis Lloyd's mirror interferometer-based laser interference lithography (LIL) system.

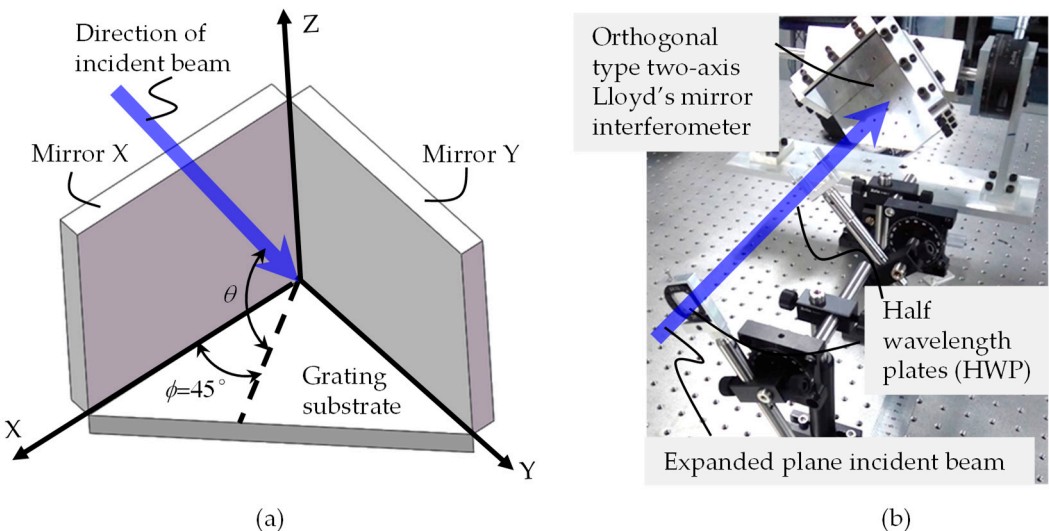

|     |     |
| :-: | :-: |
| (a) | (b) |

**Figure 2.** The global coordinate system of exposure system. (**a**) Two-axis Lloyd's mirror unit and definition of incident angle $\theta$ and azimuth angle $\varphi$. (**b**) Photograph of half-wavelength plates (HWPs) setup.

We selected the mirror with the coating of enhanced aluminum for the purpose of adjusting arbitrarily the grating pitch. The mirror with the dielectric interface, by contrast, has a limited angle of incidence, usually within a range of 0 to 45°. Two half-wavelength plates were employed in the polarization adjustment unit that controlled the polarization state of corresponding incidence beams, as shown in Figure 2b. Section 2.2 introduces the mechanism of polarization adjusting.

### 2.2. Interference Theory of Orthogonal Two-Axis Lloyd's Mirror Interferometer

According to the analysis of the interferometer, the collimated beam was divided into five sections, as shown in the Figure 3a, called beam 1, 2, 3, 4, and 5. The beam directly projecting onto the substrate is beam 1; the beam propagating onto the substrate after being reflected by the X-mirror is beam 2; the beam propagating onto the substrate after being reflected by the Y-mirror is beam 3; the beams propagating onto the substrate after being reflected by the Y-mirror firstly and reflected by the X-mirrors secondly is beam 4; and the beams propagating onto the substrate after being reflected by the X-mirror firstly and reflected by the Y-mirrors secondly is beam 5.

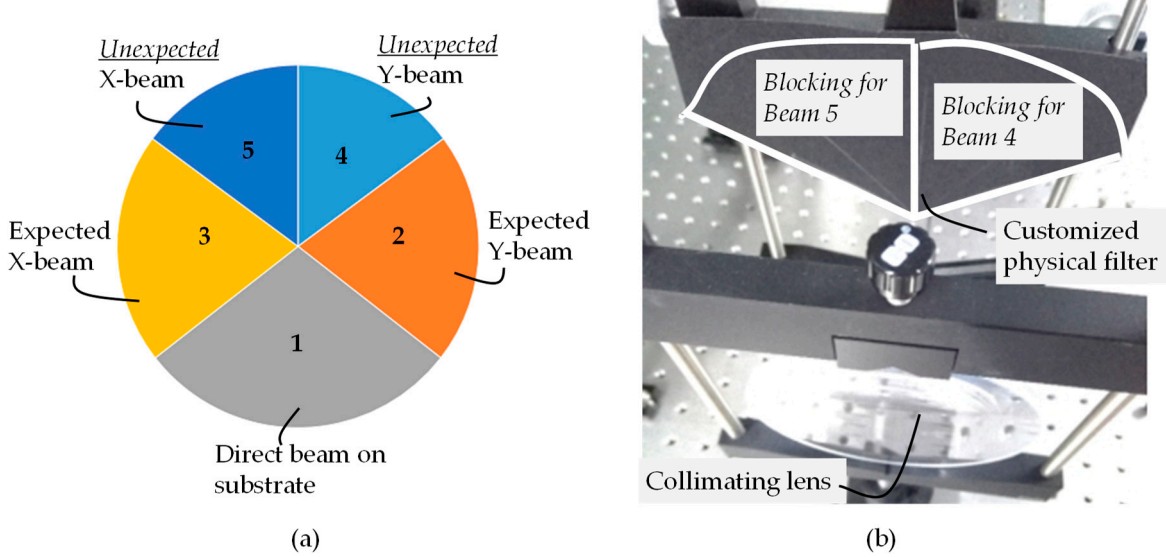

(a)             (b)

**Figure 3.** Optical configurations of orthogonal two-axis Lloyd's mirror interferometer. (**a**) The sections of collimated beam (from the direction of the incident beam). (**b**) Improved physical filter.

First of all, however, it should be noted that only beams 1, 2, and 3 are required for fabrication of the desired 2D gratings. Beams 4 and 5 will introduce unexpected grating patterns [14]. Thus, a physical filter is stuck in the optical path as shown in Figure 3b, a similar manner as that illustrated in present methods [16–18]. It should be noted that the incident plane wavefront beam still follows a Gaussian distribution, which will influence the grating shape uniformity among the whole grating surface.

Theoretically, assuming that only three—beams 1, 2, and 3—have uniform intensity distribution at the wavefront, the electric field of a laser beam can be expressed as follows.

$$\boldsymbol{E}_i = E_i \cdot \boldsymbol{e}_i exp(j\boldsymbol{k}_i \cdot \boldsymbol{r} + \gamma_i)\ (i = 1,\ 2,\ 3,\ 4,\ 5), \tag{1}$$

where $\boldsymbol{r}$ is the position vector, $\boldsymbol{E}_i$ is the real electric field amplitude, $\boldsymbol{e}_i$ is a unit vector in the polarization direction of the laser beam, $\gamma_i$ is the initial phase, and $\boldsymbol{k}_i$ is the wave vector. The total electric field of needed incident beams can be expressed by

$$\mathbf{e}(\mathbf{r}) = \sum_{i=1}^{3} E_i exp(j\boldsymbol{k}_i \cdot \boldsymbol{r} + \gamma_i)\ (i = 1,\ 2,\ 3). \tag{2}$$

The intensity distribution of interference field upon the substrate can be written as follows,

$$I(\mathbf{r}) = \sum_{i=1}^{3} |E_i|^2 + 2\sum_{m=2}^{3}\sum_{n<m} E_m E_n \boldsymbol{e}_m \boldsymbol{e}_n cos\{(\boldsymbol{k}_n - \boldsymbol{k}_m) \cdot \boldsymbol{r} + \varphi_{n-}\varphi_m\}. \tag{3}$$

According to Equation (3), the periods of the interference fringes to be generated by the interference between the two of those beams can be expressed as follows,

$$g_{nm} = \frac{2\pi}{|\mathbf{k}_n - \mathbf{k}_m|} \quad (n < m \le 3). \tag{4}$$

Assume $\varphi = 45°$ and the mirrors and substrate are orthogonal with each other, the X and Y-directional fringe periods and, respectively, can be expressed as follows,

$$g_x = g_y = \frac{\lambda}{\sqrt{2}cos\theta}. \tag{5}$$

Figure 4 summarizes the line interference fringes to be generated by the interference between the two of those beams. As we can be seen in Figure 4, the interference fringes generated by beams 1 and 2, and beams 1 and 3 are parallel to Y-axis and X-axis, respectively, as shown in Figure 4a,b. Obviously, this two interference fringes are perpendicular to each other. The interference fringe generated by beam 2 and beam 3 are 45° with respect to X-axis as shown in Figure 4c. The interference fringe is enough for fabricating a 2D-orthogonal grating with the interference fringes generated by beams 1 and 2 and beams 1 and 3. Consequently, it is necessary to eliminate the interference of beam 2 and beam 3.

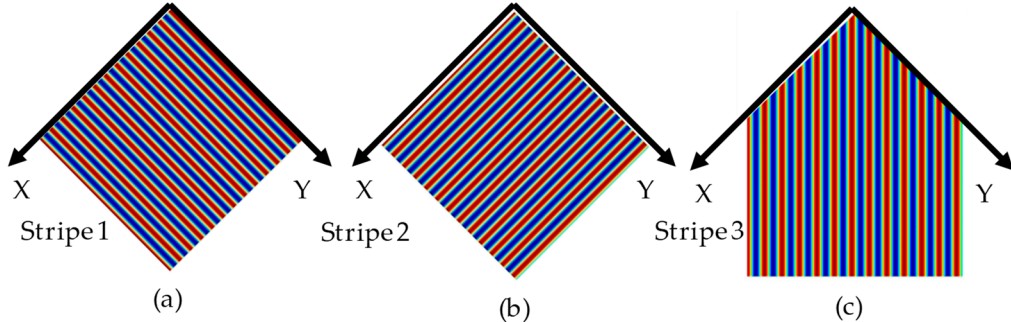

**Figure 4.** Interference fringe patterns generated by two of beam 1, 2, and 3. (**a**) beams 1 and 2; (**b**) beams 1 and 3; and (**c**) beams 2 and 3.

### 2.3. Optimum Combination of Initial Polarization

Figure 4 clearly illustrates the grating stripe direction. In previous research [13,16–18], effects are paid on elimination/reduction of Stripe 3 by using polarization modulation and experimental results verified that these proposals are effective in the reduction of Stripe 3 despite downsizing of the grating surface. But after deep investigation and careful calculation of these grating stripes with consideration of the exposure system configuration, it can be found that the influence of the Stripe 3 varies when not only the polarization but also the incident angle $\theta$ are different. As that proposed in [16], denoting the degree of orthogonality between beams 2 and 3 $\gamma_{23}$, and Stripe 3 might be neglected when $\gamma_{23}$ is lower than a threshold or an experimentally obtained tolerance.

We employed the combination of three angles ($\alpha_1$, $\alpha_2$, and $\alpha_3$) to represent the combination of the initial orientation of polarization. Counter-clockwise and clockwise rotation were recorded as positive and negative angle, respectively. With the strict method to modulate the initial polarization status based on the theory of three-dimensional polarization ray-tracing calculus [16,19], the lowest $\gamma_{23}$ of approximately 0.048 can be achieved when the incidence angle $\theta = 71.79°$ corresponding to a grating period of 1 µm under a polarization status of polarization modulated three beams (direct beam, X-beam, and Y-beam: s, $-33.5°$, $33.5°$, respectively). On the other hand, under such an incident angle, without polarization modulation, i.e., three beams polarization status (s, s, and s), the values $\gamma_{23}$ of are 0.06 and 0.08 for metal-type mirror and dielectric-type mirror, respectively. It can be seen that this difference is not significant. It can be expected that, even without polarization, the Stripe 3 will not influence the grating uniformity so much. These findings are fundamental to this research.

Unfortunately, however, it should be noted that on the premise of the incident beam with linear polarization states, there does not exist a combination of $\alpha_2$ and $\alpha_3$ that could completely prevent the interference of beam 2 and beam 3, i.e., $\gamma_{23} = 0$. For a near zero $\gamma_{23}$, polarization modulation of the incident beam might be required, this will be discussed in the future.

## 3. Experiment and Results

According to the abovementioned theories about polarization modulation, we used half-wavelength plates to realize the optimal combination of initial polarization status. We used a linear polarizer to measure the initial orientation of polarization of the laser source, and use a half-wavelength plate to modulate it with 0 degrees; we marked it as s. We employed two HWPS, which is half-wavelength plates, to alter the orientation of polarization of beams 2 and 3. However, the basic requirement is that optical elements after the collimation lens are as few as possible, to keep a good wavefront and a relatively large effective exposure area. To a certain extent, if the degree of orthogonality is small enough as mentioned above, we can ignore the interference between beam 2 and 3. Therefore, experiments with half-wavelength plates were carried out to verify the effects of polarization modulation. Meanwhile, for the purpose of obtaining maximum fabrication area and simplification of the optical system, we conducted a bold attempt that abandoned employing the two half-wavelength plates. It is worth noting that the reflected beam had different polarization status with the change of incident angle according to Fresnel's theories. For the sake of simplification, all the experiments were carried out under the constant condition of incident angle $\theta = 71.8°$. The fabricated 2D gratings could have the same grating pitch at two orthogonal directions ($g_{12} = g_{13} = 1$ μm). Figure 5 shows the fabricated gratings without and with polarization, i.e., under polarization state of (s, s, s) and (s, $-33.5°$, $33.5°$), respectively. It can be seen that the area of 2D grating fabricated without polarization status, i.e., initial polarization of the laser source (s, s, s) is larger than the modulated polarization status (s, $-33.5°$, $33.5°$). The area of the 2D grating fabricated with the initial polarization status (s, s, s) and (s, $-33.5°$, $33.5°$) are 25 mm × 25 mm and 10 mm × 10 mm, respectively. The enlarged grating area will be beneficial for grating interferometry application.

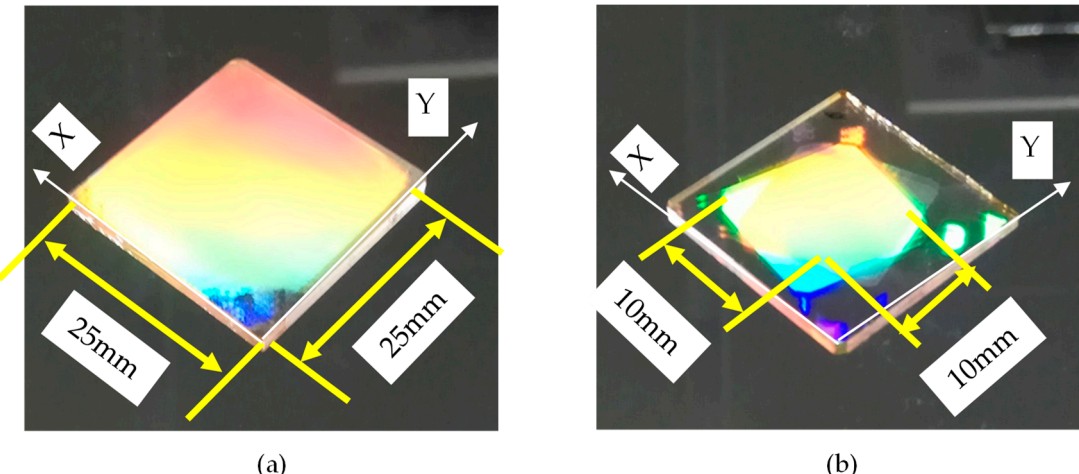

(a)                                                                         (b)

**Figure 5.** Photographs of the fabricated 2D gratings under the initial polarization status of (**a**) (s, s, s) under conditions of exposure time 30 s, development time 8 s and (**b**) (s, $-33.5°$, $33.5°$) under exposure time 30 s, development time 8 s.

The fabricated grating shapes were then evaluated by using an atomic force microscope (AFM). The AFM (Bruker Dimension Icon) worked on a peak force tapping mode with a height sensor data density of 128 × 128 of an AFM image with a size of 10 μm × 10 μm. The grating surface material is photoresist, and AFM tip material is SiC and the shape aspect ratio is 1. The peak force setpoint and amplitude are 0.05 and 300 nm, respectively. Figure 6a,b show the images of the fabricated 2D

gratings under the initial polarization status of (s, s, s) and (s, −33.5°, 33.5°), respectively. First of all, it can be seen that the grating shapes under these two polarization status are all regularly uniform, a near pillar shape, and no obvious elliptical shape as illustrated in [13] occurred. More than ten samples have been fabricated and results were highly consistent. Furthermore, the grating periods were evaluated as shown in Figure 6, B-B'. The grating periods are all consistent with the designed value of 1 µm. It should be noted that, the period consistence is more importance than the absolute grating period value. From the A-A' sectional data, we can see that the grating depth are about 500 nm, (s, s, s) is slightly lower than that of the (s, −33.5°, 33.5°), which are also consistent with the grating shape demand.

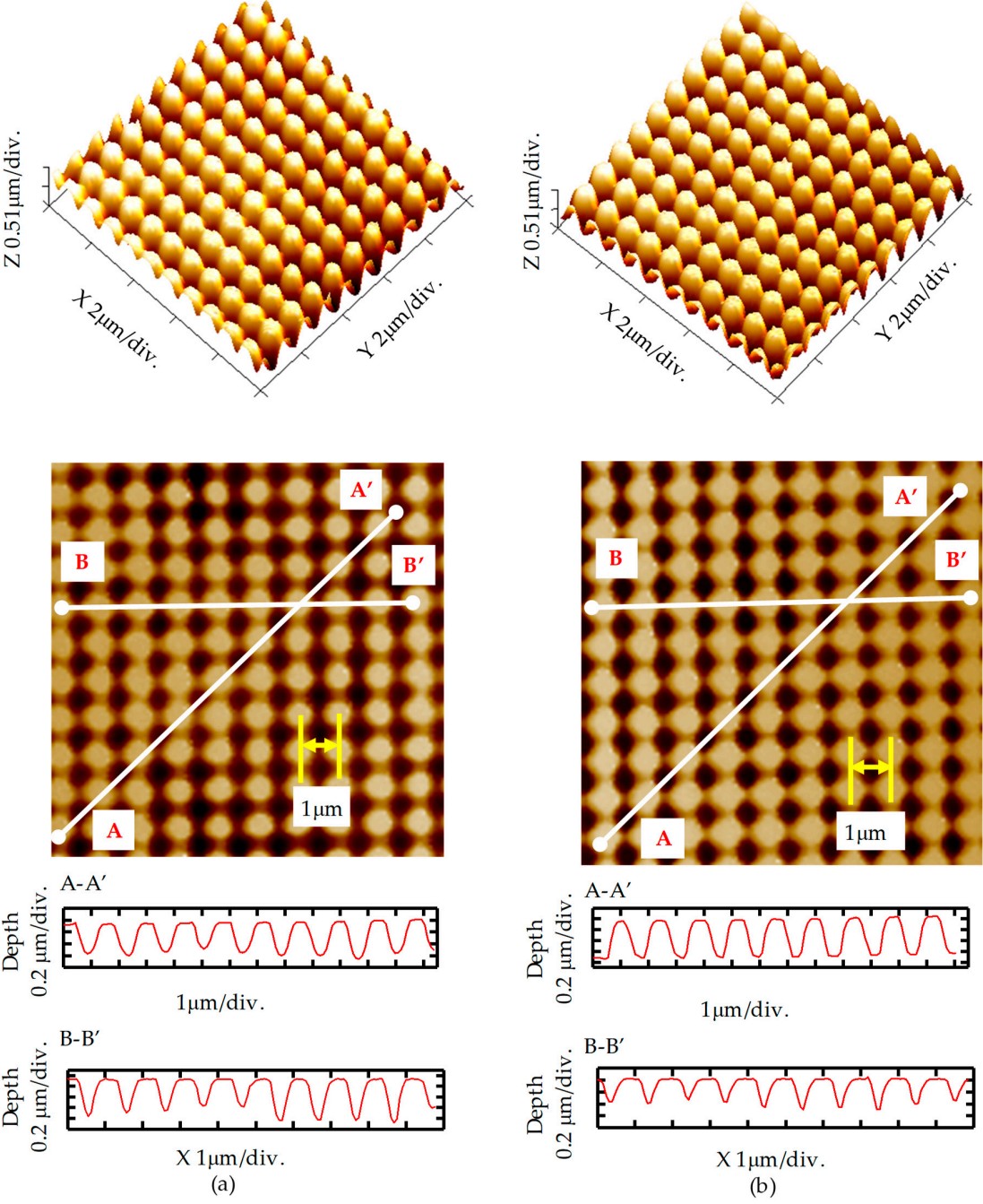

**Figure 6.** Atomic force microscopy (AFM) images and grating shape details of the fabricated 2D gratings under the initial polarization status of (**a**) (s, s, s) and (**b**) modulated polarization status (s, −33.5°, 33.5°).

From Figures 5 and 6, it can be concluded that the grating shape and area of these two polarization status are all consistent with the preliminarily designed value. However, before it can be employed to the grating interferometry system, grating diffraction efficiency that influencing the inference signals are required to be evaluated. An experimental setup, as shown in Figure 7a, was constructed and the measurement point distribution is shown in Figure 7b. As for the real interferometry system, the incident beam was polarized to be circular and pass through the fabricated grating. After the diffraction of the grating, four first-order diffracted lights appear on the observation screen. The power of each diffracted light can be read by the power meter, which was then divided by the output power of the laser source, thus we can get the diffraction efficiency. The measurement results were summarized by Tables 1 and 2. Each diffraction efficiency is larger than 6%, and all the sum efficiency of all the first diffraction beam are larger than 50%. For the uniformity of the +1 and the −1 diffraction beams along the X- and Y-directions, the initial polarization is better than the modulated polarization, this may be because the intensity was redistributed by the two QWPs. The diffraction difference between the X- and the Y-direction is mainly due to the unideal intensity uniformity within the plane wavefront.

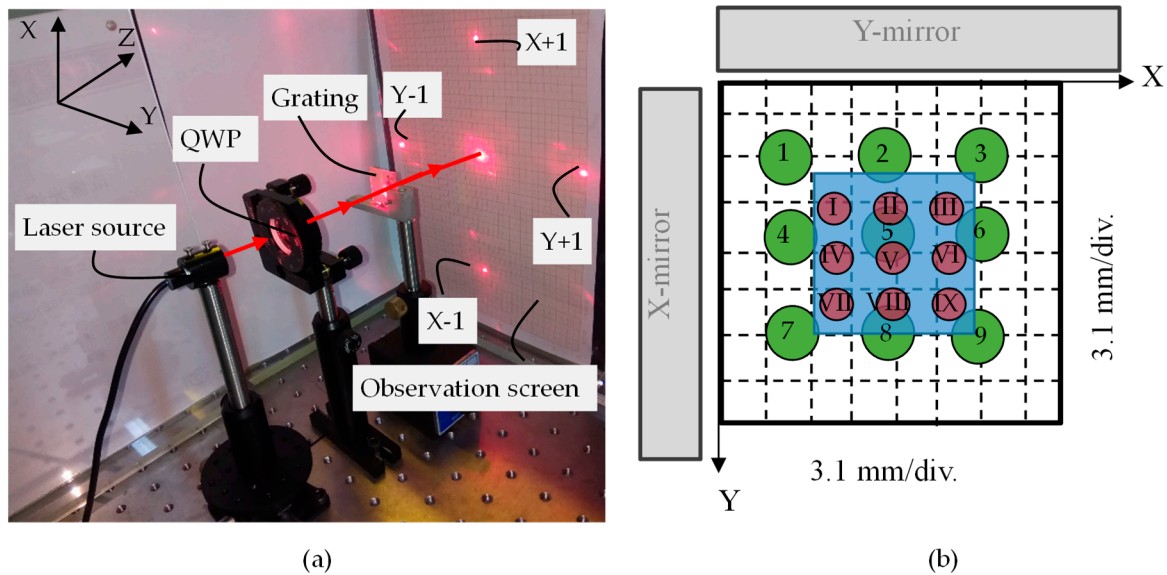

(a)                (b)

**Figure 7.** Diffraction efficiency testing setup (**a**) and measurement points on the grating surface (**b**).

**Table 1.** Diffraction efficiency testing result of the grating fabricated without polarization.

| Points | 1 | 2 | 3 | 4 | 5 | 6 | 7 | 8 | 9 |
|---|---|---|---|---|---|---|---|---|---|
| X + 1 | 0.21 | 0.18 | 0.17 | 0.22 | 0.19 | 0.19 | 0.22 | 0.18 | 0.18 |
| X − 1 | 0.19 | 0.16 | 0.16 | 0.21 | 0.18 | 0.18 | 0.21 | 0.17 | 0.17 |
| Y + 1 | 0.13 | 0.14 | 0.12 | 0.09 | 0.13 | 0.13 | 0.07 | 0.12 | 0.13 |
| Y − 1 | 0.13 | 0.14 | 0.13 | 0.08 | 0.13 | 0.13 | 0.06 | 0.13 | 0.13 |

**Table 2.** Diffraction efficiency testing result of the grating fabricated with polarization.

| Points | I | II | III | IV | V | VI | VII | VIII | IX |
|---|---|---|---|---|---|---|---|---|---|
| X + 1 | 0.06 | 0.05 | 0.06 | 0.09 | 0.06 | 0.09 | 0.14 | 0.13 | 0.12 |
| X − 1 | 0.12 | 0.08 | 0.10 | 0.17 | 0.11 | 0.14 | 0.22 | 0.19 | 0.18 |
| Y + 1 | 0.22 | 0.23 | 0.22 | 0.12 | 0.20 | 0.17 | 0.07 | 0.13 | 0.14 |
| Y − 1 | 0.18 | 0.22 | 0.19 | 0.07 | 0.17 | 0.13 | 0.01 | 0.09 | 0.12 |

For comparison, we conducted experiments at the incident angle $\theta$ of 60° and 85° without polarization modulation, i.e., (s, s, s), corresponding to the grating periods of 625 nm and 3586 nm, respectively. Under such polarization and incident angle, the $\gamma_{23}$ is larger than that of $\theta$ of 71.8°.

Figure 8 shows the AFM images. This result reveals that the groove of 2D gratings deviated from square and was more inclined to elliptical shape, which cannot meet the interferometry application. Therefore, it is necessary for fabricating 2D grating to operate HWPs at 60° and 85°, as well as other angles. The specific values of modulation need to be calculated based on the three-dimensional polarization ray-tracing calculus [20].

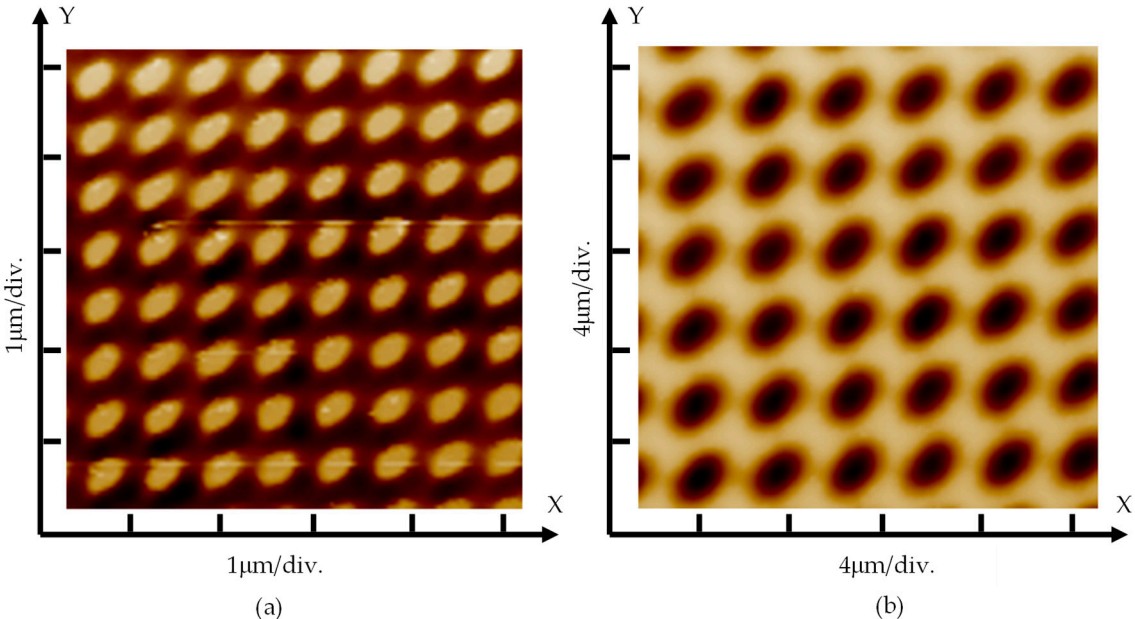

**Figure 8.** AFM images of fabricated grating without polarization under different incident angle $\theta$ shown in Figure 2: (**a**) 60° and (**b**) 85°.

## 4. Discussion and Conclusions

We have designed and assembled the orthogonal type two-axis Lloyd's mirror interferometer as the exposure system. Apart from removing of the unexpected beams 4 and 5, the grating shape uniformity is theoretically evaluated in terms of an undesired interference between beams 2 and 3 by an index of degree of orthogonality.

Experimental results reveal that the difference is not significant for the fabricated 2D gratings under the original polarization status (s, s, s) and the optimal combination polarization status (s, $-33.5°$, $33.5°$) under a specific incident angle corresponding to a 1 µm grating period in both X and Y directions. The degree of orthogonality of beams 2 and 3 that fall in the same amounts level in the above two cases is the main reason. The degree of orthogonality of beam 2 and 3 are 0.07 and 0.048, respectively. The fabrication result under polarization was consistent with that shown in previous research, where similar initial polarization states (s, $-45°$, $45°$) were employed [18]. However, the grating area would be greatly enlarged without polarization as shown in Figure 5a compared with all the simulations and experiments shown in [18]; the optical system complexity was greatly reduced compared with the present technique, TBTE [5,10].

It is also confirmed that, under the condition of incident linear polarized light, the combination of initial polarization status of beams 2 and 3 corresponding to orthogonal polarization state in the orthogonal two-axis Lloyd's interferometer does not exist. The microstructures of the fabricated gratings are all consistent with the designed value. However the diffraction efficiency uniformity, both in a relative large area and the four diffraction beams on a point, require improvement.

In summary, polarization modulation technique of the LIL system can bring a good grating shape despite grating area downsizing, while, without polarization, i.e., usage of the initial polarization under some specific angle that can reach a similar grating performance, with a reduced system complexity and an enlarge grating area.

**Author Contributions:** Conceptualization, X.L.; Methodology, X.L. and H.L.; Software, H.L.; Validation, X.L., Q.Z. and K.N.; Resources, Q.Z., G.W., and X.W.; Data Curation, H.L.; Writing—original draft preparation, X.L., H.L.; Writing—review and editing, X.L.; Visualization, X.L.; Supervision, X.L.; Project Administration, Q.Z.; Funding Acquisition, G.W. and K.N.

**Funding:** This research was supported by the National Natural Science Foundation of China under Project No. 51427805, the Youth Funding of Shenzhen Graduate of Tsinghua University with Grant No. QN20180003, Shenzhen Fundamental Research Funding Grant No. JCYJ20170817160808432 and Grant No. JCYJ20160531195459678, and the National Key Research and Development Program under Grant No. 2016YFF0100704.

**Conflicts of Interest:** The authors declare no conflict of interest.

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
