# Peer review of "An Orthogonal Type Two-Axis Lloyd’s Mirror for Holographic Fabrication of Two-Dimensional Planar Scale Gratings with Large Area"

_applsci, doi:10.3390/app8112283_

Reviewer 1 Report

The paper is well structured and it is interesting research and can be recommended for publication after some critical revisions and improvements in English.

1. Please double check in whole document in terms of English. For instance: "A novel nanofabrication technique based on 4- 61 beam interference lithography were proposed by..."

2. More details how atomic force microscope (AFM) measurements were made are needed.

3. Is it possible to use another kind of laser or even other wavelength, like 325nm nm He-Cd laser source? it mus be explained in the text for the readers have no doubts about experiments

4. In discussion and conclusion section, a discussion about the results obtained here should be compared with the literature works and highlight the advantages of this work to overcome the problems that authors verified as a motivation of this work.

5. The authors claim a interference with a order of nominal value of 0.1 but no explaination is given. Please improve some justifications.

6. Section 2.3 about optimal combination of initial polarization should be improved since it is short and confuse for a critical sub-section.

Author Response

Dear Reviewer,

Thank you very much for your careful review of our paper. 

Your precious comments and suggestions, which have greatly helped us to improve the paper, are highly appreciated.

The paper has been revised based on the comments and suggestions. The responses to the comments and questions are listed  in the attachment , a PDF file.

Thank you very much again for your careful review of our paper.

Kind regards,

Authors

Reviewer 2 Report

This manuscript describes experimentally measured results on a two-dimensional grating formation using a laser interference lithography based on orthogonal-type two-axis Lloyd’s mirror interferometer without polarization modulation. The reported results may be interesting to readers this technical area. I would recommend this manuscript for publication in Applied Science after some minor improvements are made on the following points:

- In line number 75 of page 2, correct the spelling of the word “patters”.

- In line number 164 of page 5, the parameters alpha2 and alpha3 were used without any explanation. Authors should include a detailed description of those parameters.

- In line numbers from 180 to 184 of page 6, no description how the polarization status was measured when the 2D gratings were fabricated. Authors may include how the polarization status of the three beams were measured at the grating formation area.

- In line number 197 of page 6, the grating periods are all consistent with the designed value, I micrometer. However, the results reported in Fig. 6(a) show different grating periods for either A-A’ section and B-B’ section while the grating period for the B-B’ section is about 1 micrometer. Authors should check this matter carefully and provide correct description to express exact results.

- In line number 210 of page 7 and Tables 1 and 2 in page 8, there is no description how the diffraction efficiencies are determined. Authors need to include some description about how the diffraction efficiencies are measured.

- In line numbers 222 to 223 of page 8, the grating periods of 625 nm and 3586 nm for the incident angles of 60 and 85 degrees, respectively. It is not clear that these grating periods were taken from the A-A’ section or from the B-B’ section as shown in Fig. 6. Authors should clarify this point.

- In line numbers from 224 to 226 of page 8, the sentence “This result reveals that the groove of 2D gratings …… ” is not clear in English. Authors should check the sentence.

- In line numbers 239-240 of page 9, the meaning of “… a 1 um grating period in both two 239 direction” is not clear. What do the authors mean for both directions? Authors should provide exact expression(s) to describe s a clear physical meaning in appropriate ways.

- In Section 4 Discussion and conclusions of page 9, the words “the degree of orthogonal” may be rewritten as ““the degree of orthogonality”.

- In line number 247 of page 9, do the words “In sum, …” mean “In summary, …”?

- Correction of some grammatical errors and inappropriate expressions in English is needed at several places of the manuscript.

End

Author Response

Dear Reviewer,

Thank you very much for your careful review of our paper. 

Your precious comments and suggestions, which have greatly helped us to improve the paper, are highly appreciated.

The paper has been revised based on the comments and suggestions. The responses to the comments and questions are listed  in the attachment , a PDF file.

Thank you very much again for your careful review of our paper.

Kind regards,

Authors

Round  2

Reviewer 1 Report

I am happy with the changes.